# Capturing Gaze Shifts for Guidance: Cross-Modal Fusion Enhancement for VLM Hallucination Mitigation

## Abstract

Vision language models (VLMs) often generate hallucination, i.e., content that cannot be substantiated by either textual or visual inputs. Prior work primarily attributes this to over-reliance on linguistic prior knowledge rather than visual inputs. Some methods attempt to mitigate hallucination by amplifying visual token attention proportionally to their attention scores. However, these methods overlook the visual attention sink problem, where attention is frequently misallocated to task-irrelevant visual regions, and neglect cross-modal fusion balance by enhancing only visual attention without adjusting attention to the user query. This can result in amplifying incorrect areas while failing to properly interpret the user query. To address these challenges, we propose a simple yet effective method called **G**aze Sh**i**ft-Guided Cross-modal **F**usion Enhancemen**t** (**GIFT**). GIFT precomputes a holistic visual saliency map by tracking positive changes in visual attention, or *"gaze shifts"*, during user query comprehension, and leverages this map to amplify attention to both salient visual information and the user query at each decoding step. This reduces the impact of visual attention sink, as irrelevant tokens exhibit minimal shifts, while ensuring balanced cross-modal fusion for well-integrated representation. Extensive experiments show that GIFT effectively mitigates hallucination in VLMs across both generative and classification tasks, achieving up to 20.7% improvement over greedy decoding, while maintaining general vision-language performance with low computational overhead.

## 1 Introduction

Vision language models (VLMs) (Li et al., 2023c; Liu et al., 2023b; Zhu et al., 2023; Liu et al., 2024b; Hurst et al., 2024; Wang et al., 2024b; Bai et al., 2025) have recently achieved remarkable progress on tasks that require joint reasoning over textual and visual information, such as visual question answering, visual reasoning, and image captioning. Despite these advances, VLMs remain prone to generating hallucination, i.e., content that cannot be substantiated by either textual or visual inputs (Liu et al., 2024a). This issue poses serious challenges, particularly in high-stakes domains such as biomedicine (Li et al., 2023b; Chen et al., 2024b), autonomous driving (Wang et al., 2023; Li et al., 2025), and robotics (Chen et al., 2024a; Li et al., 2024), where factual accuracy and reliability are critical for safe and effective operation.

Recent analyses suggest that these failures are primarily due to vision language models (VLMs) over-relying on linguistic prior knowledge while under-utilizing visual inputs (Wang et al., 2024a; Zhang et al., 2024). To mitigate this, inference-time interventions have been proposed to enhance visual grounding by highlighting visual signals based on visual saliency, i.e., the relevance of specific visual regions to the task at hand. For instance, Yin et al. (2025) enhances attention allocated to visual tokens during decoding in proportion to their attention scores. While this approach strengthens the contribution of visual information, it does not account for the balance between visual and query signals during cross-modal fusion. Consequently, the model may attend to relevant regions but misinterpret the query, forming inaccurate integrated representations. Moreover, this approach does not address the issue of visual attention sink (Kang et al., 2025), where attention is persistently misallocated to irrelevant visual tokens, potentially amplifying incorrect regions throughout generation. To mitigate the issue of visual attention sink, existing methods recalibrate visual token

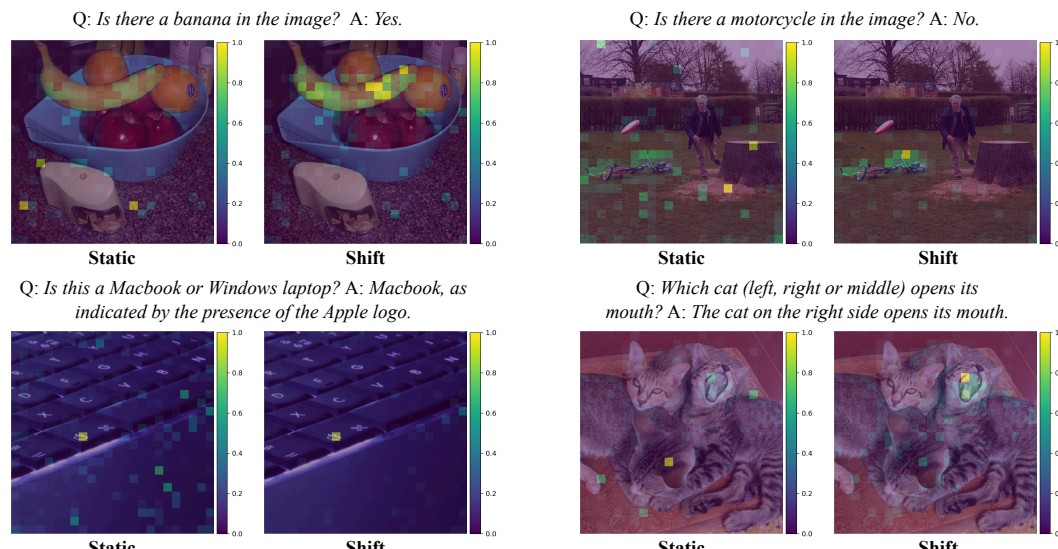

Figure 1: **Examples of visual saliency maps from LLaVA 1.5 7B.** The vanilla method (**Static**) averages visual token attention over all query tokens, while the proposed method (**Shift**) averages positive changes in visual token attention over information-rich query tokens. **Shift** more effectively highlights task-relevant visual regions and mitigates the impact of visual attention sink, where irrelevant regions receive erroneously high saliency scores.

attention (Kang et al., 2025; Zhu et al., 2025b), but they fail to address the broader problem of low overall visual contribution during decoding.

To address these limitations, we propose an inference-time hallucination mitigation method for VLMs called **G**aze Sh**i**ft-Guided Cross-modal **F**usion Enhancemen**t** (**GIFT**). Drawing inspiration from human vision, we hypothesize that VLMs, like humans, dynamically shift their "visual gaze" when processing information-rich words in a user query. By tracking positive changes in visual token attention, i.e., "gaze shifts", over these information-rich query tokens at the layer exhibiting the largest positive change, GIFT pre-computes a holistic visual saliency map that captures task-relevant regions prior to decoding, requiring pre-filling only up to that layer. This mechanism also mitigates the impact of visual attention sink, as irrelevant regions exhibit minimal or no attention shift. During decoding, GIFT leverages this saliency map to proportionally amplify attention to salient visual tokens in critical cross-modal fusion layers, where the model attends strongly to both visual and query tokens. In contrast to Yin et al. (2025), which only increases attention to visual tokens, GIFT also adjusts attention to query tokens based on the overall visual attention amplification ratio, maintaining cross-modal balance and forming well-integrated representations.

In summary, our main contributions are three-fold:

- We introduce a novel mechanism that captures a holistic view of salient visual regions while effectively mitigating the visual attention sink problem. This mechanism pre-computes a task-relevant visual saliency map prior to decoding by tracking positive shifts in visual attention, i.e., "gaze shifts", as the VLM processes information-rich words in a user query.

- We propose GIFT, a lightweight inference-time hallucination mitigation method that leverages the precomputed saliency map to guide visual attention enhancement while proportionally scaling attention to query tokens to preserve cross-modal fusion balance.

- We show that GIFT consistently mitigates hallucination across VLM architectures and model sizes, achieving gains of up to 20.7% on CHAIR, 15.9% on MMHal-Bench, and 3.0% improvement on POPE, while preserving general vision-language performance with low computational overhead. Extensive ablation studies further validate the contribution of each component.

## 2 RELATED WORK

**VLM Hallucination Mitigation.** A key cause of hallucination in VLMs is over-reliance on linguistic prior knowledge rather than visual inputs (Wang et al., 2024a; Zhang et al., 2024). To address this, training-based approaches have introduced specialized learnable modules (Zhao et al., 2024) or curated data augmentations (Liu et al., 2023a; Pi et al., 2024; Chen et al., 2025) to encourage stronger reliance on visual features. While effective, these methods often suffer from high computational costs and limited scalability.

Another line of work focuses on inference-time mitigation, which can be broadly categorized into three types: (1) Contrastive decoding (Leng et al., 2024; Liu et al., 2024c; Huo et al., 2024; Wang et al., 2025; Zhu et al., 2025a), which reduces over-reliance on knowledge priors by contrasting the output distributions of two inputs, one with the original visual inputs and one with perturbed or absent visual inputs. However, this approach incurs significant computational overhead due to the need for generating counterpart outputs. Our method instead strengthens visual contributions directly in intermediate layers, eliminating the need for generating alternatives. (2) Visual input modification, which manipulates the raw image to emphasize salient regions derived from intermediate signals such as visual attention, by blurring irrelevant areas (Yu et al., 2024), magnifying key regions (Mao et al., 2025), or cropping salient patches (Zhang et al., 2025). These techniques typically require additional forward passes or auxiliary inputs, increasing computational cost, and can struggle when multiple regions are salient or when a single region is overly large. In contrast, our method operates directly on intermediate outputs without constraints on the number or size of relevant regions. (3) Attention steering, which directly amplifies attention towards visual tokens, either by applying a constant value (Zhu et al., 2025a) or scaling proportionally to attention scores (Yin et al., 2025). While this increases the contribution of visual features, such methods often neglect cross-modal fusion balance, i.e., overemphasizing visual features without adequately reinforcing query token attention can impair proper comprehension. They also overlook the visual attention sink problem (Kang et al., 2025), which our approach explicitly addresses.

**Attention Sink.** Attention sink refers to the phenomenon where task-irrelevant sink tokens, such as those with limited semantic meaning or representing background, receive disproportionately high attention weights. This issue has been observed in both language models (Xiao et al., 2024; Ferrando & Voita, 2024) and vision transformers (Darcet et al., 2023). Similar patterns have been identified in vision language models (VLMs) by Kang et al. (2025) and Zhu et al. (2025b), where mitigation strategies recalibrate visual attention to suppress these sink tokens. However, these methods do not address a broader limitation that visual features contribute relatively little during generation. In this work, we present a simple yet effective method that pre-computes a task-relevant visual saliency map by tracking positive changes in visual attention, i.e., "gaze shifts", over information-rich query tokens. During decoding, this saliency map, which is robust to visual attention sink, is used to jointly amplify attention to both visual and query tokens, improving cross-modal integration.

## 3 VISUAL SALIENCY MAP COMPUTATION VIA GAZE SHIFT TRACKING

In this section, we present our mechanism for computing a visual saliency map that captures a holistic view of salient visual regions while mitigating the visual attention sink problem. We first examine whether a simple average of visual attention across user query tokens, referred to as **"static gaze"**, can effectively produce a holistic, noise-free visual saliency map.

Vision language models (VLMs) typically process three inputs: a system instruction $s$, visual inputs $v$, and a user text query $t$. The system instruction and query are tokenized into sequences $X_S$ and $X_T$, while the visual input $v$ is encoded by a visual encoder into dense embeddings and then projected into text-aligned visual tokens $X_V$ (Liu et al., 2023b; Wang et al., 2024b). These components are concatenated as $X = [X_S; X_V; X_T]$, and passed into a large language model (LLM) to generate output tokens autoregressively:

$$y_t = \arg\max p_\theta \left( y_t \mid y_{<t}, X_S, X_V, X_T \right) \tag{1}$$

where $y_{<t}$ denotes the sequence of previously generated tokens.

Within the model, the attention matrix $\boldsymbol{A}^l \in \mathbb{R}^{h \times n \times n}$ encodes how each of the $n$ tokens attends to all others across $h$ attention heads at layer $l$. For simplicity, batch dimensions are omitted. Visual

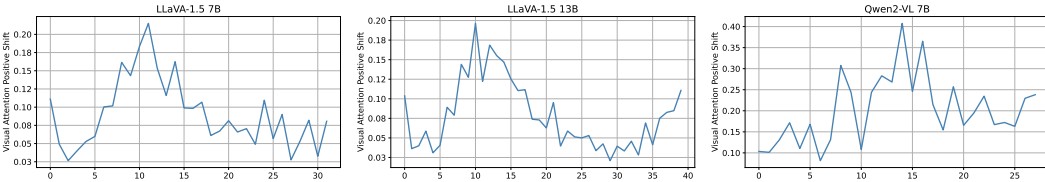

Figure 2: **Volume of visual attention positive shifts in VLMs when processing information-rich query tokens across layers.** The volume reflects how strongly the model reallocates focus within visual regions, with the largest shifts occurring in early to middle layers, indicating that VLMs settle on relevant visual regions once identified.

attention sink (Kang et al., 2025) refers to the phenomenon where query-irrelevant visual tokens receive disproportionately high attention. From this matrix, we extract the submatrix representing attention from query tokens $X_T$ to visual tokens $X_V$, which serves as the foundation for constructing the visual saliency map.

Prior work (He et al., 2024; Yin et al., 2025; Kang et al., 2025) has demonstrated that only a subset of attention heads are primarily responsible for attending to visual information. Following their findings, at each layer $l$, we select the top 50% of attention heads with the highest cumulative attention to visual tokens $X_V$ aggregated across all query tokens $X_T$, denoted as $\mathcal{H}_{TV}^l$. We then compute the mean attention over these selected heads and average across all query tokens, applying min-max normalization to produce the saliency map $\mathcal{S}^l$:

$$\mathcal{S}^l = \text{Min-max}\left(\frac{1}{|\mathcal{H}_{TV}^l| \cdot |X_T|} \sum_{h \in \mathcal{H}_{TV}^l} \sum_{i \in X_T} \mathbf{A}_{h,i,j}^l\right), \quad j \in X_V \tag{2}$$

where $h$ indexes attention heads $\mathcal{H}_{TV}^l$, $i$ indexes query tokens $X_T$, and $j$ indexes visual tokens $X_V$.

Figure 1 shows "static" saliency maps from LLaVA-1.5 7B (Liu et al., 2023b) on the left side of each example. While they partially highlight relevant visual regions, they often assign high saliency scores to irrelevant areas as well. This misallocation, known as visual attention sink (Kang et al., 2025), can produce misleading signals, such as emphasizing unrelated features, and negatively affect downstream generation.

To address this, we propose a simple yet effective approach inspired by human vision. We hypothesize that, like humans, VLMs dynamically shift their "visual gaze" to capture relevant visual information while comprehending the user query. By tracking positive changes in visual attention, referred to as **"gaze shifts"**, over information-rich query tokens, we obtain a holistic view of task-relevant visual regions. Since irrelevant regions typically exhibit minimal or no change in attention, this approach naturally mitigates the issue of visual attention sink. We restrict tracking to information-rich words, where attention shifts are most meaningful, and consider only positive shifts, since negative shifts merely indicate moving focus away from previously salient regions and would cancel out meaningful increases.

Concretely, we first extract information-rich words from the user query using spaCy's Part-Of-Speech (POS) tagging (Honnibal et al., 2020), which incurs minimal computational overhead, selecting words tagged as NOUN, PROPN, VERB, ADJ, ADV, or NUM. These words correspond to a set of query tokens $X_{Tr}$. At layer $l$, we select the top 50% of attention heads, $\hat{\mathcal{H}}_{TrV}^l$, with the highest cumulative positive changes in attention to visual tokens aggregated across these information-rich query tokens. Here, the positive change in attention is defined as the increase in visual attention from the previous query to the current one, with negative changes set to zero. Using these heads and query tokens, we compute a refined saliency map that captures the average positive shift in visual attention, emphasizing task-relevant regions:

$$\hat{\mathcal{S}}^l = \text{Min-max}\left(\frac{1}{|\hat{\mathcal{H}}_{TrV}^l| \cdot |X_{Tr}|} \sum_{h \in \hat{\mathcal{H}}_{TrV}^l} \sum_{i \in X_{Tr}} \max(\mathbf{A}_{h,i,j}^l - \mathbf{A}_{h,i-1,j}^l, 0)\right), \quad j \in X_V \tag{3}$$

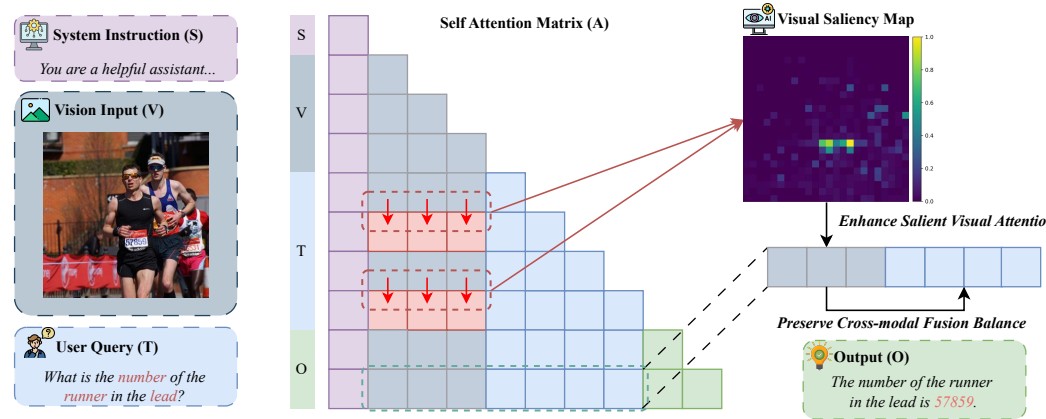

Figure 3: **Illustration of GIFT.** GIFT tracks positive changes in visual attention, i.e., "gaze shifts" across information-rich query tokens to compute a visual saliency map, which guides enhancement of salient visual attention while preserving cross-modal fusion balance.

Figure 1 shows the resulting "shift" saliency maps from LLaVA-1.5 7B on the right side of each example, which more accurately highlight relevant regions and reduce noise from irrelevant areas compared to "static" maps.

To select the optimal layer for computing the visual saliency map, we sample 50 examples from the training set of TextVQA (Singh et al., 2019), a visual question answering dataset, and measure the sum of $\hat{S}^l$ before min-max normalization to identify where visual attention is most dynamically realigned during query processing. As shown in Figure 2, this peak generally occurs in the early to middle layers across models. In the following sections, we denote the visual saliency map extracted from the optimal layer as $\hat{S}$.

We quantitatively compare the "static" and "shift" approaches using 1,000 examples from the MSCOCO 2014 training set (Lin et al., 2014), each consisting of an image, an object instance, and its bounding box. The "static" maps are obtained by averaging visual attention across all query tokens, while the "shift" maps track positive changes in visual attention across information-rich query tokens. To avoid confounding factors, we restrict the examples to instances whose category appears only once in the image. For evaluation, we replace min-max normalization with sum normalization so that each saliency map sums to 1, and measure the fraction of total saliency falling inside the bounding box, normalized by the box's area relative to the image. Table 1 shows that the "shift" method achieves significantly higher scores, indicating stronger focus on task-relevant regions and reduced noise from irrelevant areas.

Table 1: **Comparison of visual saliency methods.** The score represents the proportion of saliency falling inside the bounding box, normalized by the box's relative area.

|  | Static | Shift |
| --- | --- | --- |
| Norm. Saliency Score | 5.40 | **11.92** |

## 4  GAZE SHIFT-GUIDED CROSS-MODAL FUSION ENHANCEMENT

Having computed a holistic, noise-reduced visual saliency map, we introduce our hallucination mitigation method, illustrated in Figure 3, which leverages it to guide cross-modal fusion enhancement through attention steering during decoding.

**Selecting Cross-modal Fusion Enhancement Layers.** We first examine which layers are most effective for enhancing cross-modal fusion. Unlike the visual saliency map in Section 3, which tracks attention flow from query tokens to visual tokens, here we analyze flow from output tokens to both query and visual tokens. Using the 50 TextVQA examples, we measure the proportion of attention allocated to visual tokens and to query tokens over information-rich output words $Y_r$, denoted as $\mathcal{R}_V^l$ and $\mathcal{R}_T^l$, and at each layer retain the top 50% attention heads with the highest values,

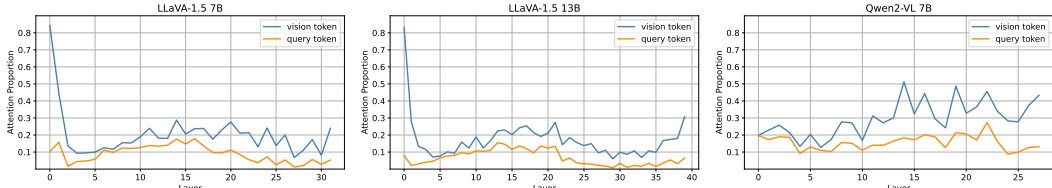

Figure 4: **Attention proportions of query and visual tokens from output tokens across layers.** The attention proportions of query and visual tokens follow similar patterns across layers, suggesting that effective cross-modal fusion relies on contributions from both modalities.

denoted as $\mathcal{H}_{OV}^l$ and $\mathcal{H}_{OT}^l$. To ensure attention patterns are not dominated by linguistic priors, which intensify with output length (Min et al., 2024; Xie et al., 2025), we restrict outputs to a single sentence. Formally:

$$\mathcal{R}_V^l = \frac{1}{|\mathcal{H}_{OV}^l| \cdot |Y_r|} \sum_{h \in \mathcal{H}_{OV}^l} \sum_{i \in Y_r} \sum_{j \in X_V} \boldsymbol{A}_{h,i,j}^l, \quad \mathcal{R}_T^l = \frac{1}{|\mathcal{H}_{OT}^l| \cdot |Y_r|} \sum_{h \in \mathcal{H}_{OT}^l} \sum_{i \in Y_r} \sum_{j \in X_T} \boldsymbol{A}_{h,i,j}^l \quad (4)$$

Figure 4 shows a consistent trend across models that attention proportions to query and visual tokens rise and fall together across layers, highlighting their joint contribution to well-integrated representations, though the absolute attention levels remain low. This indicates that effective cross-modal fusion requires simultaneously enhancing attention to both. We select layers with high attention to both visual and query tokens for enhancement, denoted as $\mathcal{L}$, as these are the layers where cross-modal fusion is most active.

**Enhancing Visual Attention via Attention Steering.** Attention steering (Zhang et al., 2023) aims to bias the attention matrix $\boldsymbol{A}$ toward salient tokens by adding a learned or heuristic bias $\boldsymbol{B}$:

$$\boldsymbol{A} = \text{Softmax}\left(\frac{\boldsymbol{Q} \cdot \boldsymbol{K}^\top}{\sqrt{d_k}} + \boldsymbol{B} + M\right) \quad (5)$$

where $\boldsymbol{B}$ assigns positive values only to salient tokens at specific attention heads and $M$ is attention mask. In VLMs, these salient tokens typically correspond to visual tokens representing important image regions. Prior approaches either apply constant biases (Zhu et al., 2025a) or scale them proportionally to attention scores (Yin et al., 2025) at each decoding step.

In our work, for the selected layers $\mathcal{L}$, we enhance visual token attention for the top heads $\mathcal{H}_{OV}^l$ at each decoding step using the pre-computed saliency map. Unlike Yin et al. (2025), which relies on attention scores at the current step, our map is derived from user query processing, providing a holistic view of visual saliency with full query context. The formulation is defined as:

$$\hat{\boldsymbol{A}}_{h,-1,j}^l = \boldsymbol{A}_{h,-1,j}^l \cdot \exp(\alpha \hat{\mathcal{S}}_j), \quad l \in \mathcal{L}, \ h \in \mathcal{H}_{OV}^l, \ j \in X_V \quad (6)$$

where $\hat{\mathcal{S}}_j$ is the saliency score for visual token $j$, $\alpha$ is a scaling factor, and $-1$ denotes the current decoding position. After sum normalization, this is equivalent to Eq. 5 with bias term $\boldsymbol{B} = \alpha \hat{\mathcal{S}}_j$. To reduce the impact of outliers, we clip the saliency map $\hat{\mathcal{S}}$ in Eq. 3 at three standard deviations before min-max normalization, preventing overemphasis on any single region.

**Balancing Cross-modal Fusion.** Previous attention steering approaches (Yin et al., 2025; Zhu et al., 2025a) focus solely on visual tokens, neglecting the contribution of query tokens in cross-modal fusion. As shown in Figure 4, the attention proportions of query and visual tokens tend to move together across layers, and both remain low even in layers with relatively higher proportions. Boosting only visual attention may improve grounding, but it risks weakening query comprehension, which is crucial for properly intepreting and utilizing visual information.

Table 2: **Performance on vision-hallucination datasets.** Our method, GIFT, outperforms all baselines across datasets and models. The best results are highlighted in bold.

| Model | Method | CHAIR | | POPE | | MMHal-Bench | |
|---|---|---|---|---|---|---|---|
| | | $C_s$ ($\downarrow$) | $C_i$ ($\downarrow$) | F1 ($\uparrow$) | Acc. ($\uparrow$) | Hal. ($\downarrow$) | Score ($\uparrow$) |
| LLaVA-1.5 7B | Greedy | 50.2 | 15.4 | 82.4 | 79.5 | 65.2 | 2.22 |
| | VAF | 49.6 | 14.3 | 81.0 | 77.2 | 66.3 | 2.16 |
| | Rel-Attn | 49.0 | 13.6 | 82.0 | 78.3 | 63.7 | 2.19 |
| | VAR | 54.0 | 15.5 | 83.1 | 80.1 | 60.8 | 2.40 |
| | **Ours** | **39.8** | **10.6** | **83.8** | **81.9** | **57.3** | **2.48** |
| LLaVA-1.5 13B | Greedy | 46.8 | 13.1 | 81.7 | 78.2 | 56.2 | 2.61 |
| | VAF | 47.4 | 13.2 | 80.6 | 76.4 | 59.2 | 2.46 |
| | Rel-Attn | 44.6 | 13.2 | 81.5 | 77.8 | 65.6 | 2.15 |
| | VAR | 51.8 | 14.0 | **82.2** | 78.5 | 56.2 | 2.52 |
| | **Ours** | **39.6** | **11.9** | 82.1 | **78.9** | **55.8** | **2.72** |
| Qwen2-VL 7B | Greedy | 24.8 | 9.1 | 86.0 | 86.5 | 32.7 | 3.53 |
| | **Ours** | **21.2** | **7.7** | **86.8** | **86.9** | **27.5** | **3.58** |

To preserve the balance of cross-modal fusion, we also scale query token attention proportionally to the overall visual attention enhancement. Formally:

$$\hat{\boldsymbol{A}}_{h,-1,j}^l = \boldsymbol{A}_{h,-1,j}^l \cdot \beta r^l, \quad l \in \mathcal{L},\ h \in \mathcal{H}_{OT}^l,\ j \in X_T$$

$$r^l = \sum_{h \in \mathcal{H}_{OV}^l} \sum_{j \in X_V} \frac{\hat{\boldsymbol{A}}_{h,-1,j}^l}{\boldsymbol{A}_{h,-1,j}^l} \tag{7}$$

where $r^l$ quantifies the overall relative increase in visual attention at layer $l$, and $\beta$ is a scaling coefficient. After scaling, we normalize the enhanced attention matrix so that, for each head and position, the attention across all tokens sums to one: $\hat{\boldsymbol{A}}_{h,i,:}^l \leftarrow \hat{\boldsymbol{A}}_{h,i,:}^l / \sum_j \hat{\boldsymbol{A}}_{h,i,j}^l$.

## 5 EXPERIMENTS

### 5.1 EXPERIMENTAL SETUP

**Models and Baselines.** We evaluate our method on three models of varying architectures and sizes, including LLaVA-1.5 7B (Liu et al., 2023b), LLaVA-1.5 13B (Liu et al., 2023b), and Qwen2-VL 7B (Wang et al., 2024b). We compare it against standard greedy decoding and three closely related approaches. VAF (Yin et al., 2025) amplifies the contribution of visual tokens by scaling visual token attention in proportion to their attention scores at each decoding step. VAR (Kang et al., 2025) identifies visual sink tokens based on model-specific hidden state dimensions and redistributes their attention proportionally to non-sink tokens in "image-centric" attention heads. MLLMs_know (Zhang et al., 2025) crops salient visual regions as additional inputs to the model, using different strategies to identify salient regions; we evaluate the best-performing variant, Rel-Attn. Since these three approaches do not provide Qwen2-VL implementations or configurations, we compare our method with them only using the LLaVA-1.5 7B and 13B models.

**Benchmark and Metrics.** We evaluate our method on both vision-hallucination datasets and general vision-language task datasets to assess its effectiveness in reducing hallucination while maintaining reasoning capabilities, as overemphasizing visual perception can potentially impair reasoning. The vision-hallucination datasets cover three tasks: POPE (Li et al., 2023d) for object detection, evaluated using F1 and accuracy; CHAIR (Rohrbach et al., 2018) for image captioning, evaluated using CHAIRs and CHAIRi; and MMHal-Bench (Sun et al., 2023) for vision question answering, evaluated using hallucination rate and informativeness score. The general vision-language datasets include MME (Fu et al., 2023) and SEED-Bench (Li et al., 2023a), both evaluated using accuracy. Further details on datasets and metrics are provided in Appendix A.

**Implementation Details.** We set $\alpha$ to 5.0 for LLaVA-1.5 7B and 13B, and 4.0 for Qwen2-VL 7B. The higher value for LLaVA, determined via hyperparameter tuning, reflects its lower original visual and query token attention compared to Qwen, as shown in Figure 4. Visual saliency maps are computed at layer 11 for LLaVA-1.5 7B, 10 for LLaVA-1.5 13B, and 14 for Qwen2-VL 7B, while cross-modal fusion is enhanced at layers 12-22, 14-20, and 5-18, respectively. We set $\beta$ to 1.0 for all models to preserve the original cross-modal balance. Additional implementation and hyperparameter tuning details are provided in Appendix B and Appendix C, respectively.

## 5.2 EXPERIMENTAL RESULTS

Table 2 presents the performance on vision-hallucination datasets. GIFT consistently outperforms all baselines across datasets, including CHAIR, POPE, and MMHal-Bench, and across models of varying architectures and sizes, including LLaVA-1.5 7B, LLaVA-1.5 13B, and Qwen2-VL 7B. Compared to greedy decoding, GIFT achieves improvements of up to 20.7% on CHAIR, 15.9% on MMHal-Bench, and 3.0% on POPE, while also improving output informativeness on MMHal-Bench by 11.7%. These results demonstrate its effectiveness and robustness in mitigating hallucinations in vision-language models across diverse evaluation settings. Qualitative examples from MMHal-Bench are provided in Appendix E to further illustrate its impact.

Table 3: **Results on general vision-language datasets.**

| Model | Method | SEED. | MME |
|---|---|---|---|
| | Greedy | 65.5 | 1751.6 |
| | VAF | 64.9 | 1787.6 |
| LLaVA-1.5 7B | Rel-Attn | 65.0 | 1811.3 |
| | VAR | 65.4 | 1780.8 |
| | Ours | 65.6 | 1750.5 |
| | Greedy | 67.8 | 1807.5 |
| | VAF | 67.0 | 1815.4 |
| LLaVA-1.5 13B | Rel-Attn | 66.7 | 1758.5 |
| | VAR | 67.9 | 1782.5 |
| | Ours | 67.7 | 1815.9 |
| Qwen2-VL 7B | Greedy | 76.0 | 2278.9 |
| | Ours | 76.0 | 2279.1 |

In addition, Table 3 presents benchmarking results on two general vision-language task datasets, SEED-Bench and MME. GIFT consistently achieves performance comparable to greedy decoding across datasets and models, indicating minimal impact on reasoning capabilities. In contrast, other baseline methods exhibit mixed results, highlighting the challenge of mitigating hallucination without compromising reasoning.

## 6 ANALYSES

In this section, we analyze GIFT's performance from three perspectives: the contribution of visual attention enhancement and cross-modal fusion balance, the impact of the enhancement coefficient $\alpha$ on hallucination and reasoning, and its computational efficiency compared to baselines.

**Cross-modal Fusion Enhancement.** We perform ablation studies to evaluate the joint contribution of two components: (1) enhancing visual token attention in proportion to task-relevant saliency, and (2) preserving cross-modal fusion balance. We consider two variants: one that increases visual attention without maintaining fusion balance by omitting Eq.7, and another that calibrates visual attention distribution to emphasize salient tokens while keeping the overall visual contribution unchanged. In the latter, the enhanced visual attention $\hat{\mathcal{A}}^l$ is scaled down by $r^l$ from Eq.7, leaving query token attention unchanged. Evaluation is conducted on two vision-hallucination datasets, POPE and MMHal-Bench. We exclude CHAIR as its image captioning queries lack sufficient specificity for query attention enhancement to meaningfully improve cross-modal representations. Table 4 shows that our full method consistently outperforms both ablation variants across datasets and models by up to 21.9%, underscoring that both components are essential for effective hallucination mitigation.

**Impact of Enhancement Coefficient.** To examine the impact of enhancement coefficient $\alpha$ on hallucination and general reasoning capabilities, we evaluate our method with $\alpha$ values ranging from 1.0 to 7.0 on the POPE and MME datasets. Figure 5 presents the results for LLaVA-1.5 7B. As $\alpha$ increases, the hallucination rate steadily decreases, as indicated by the POPE curve. However, when $\alpha$ exceeds 5.0, performance on the MME dataset drops below that of greedy decoding, suggesting that excessive emphasis on visual and query tokens may lead the model to overfit to perceptual details while underutilizing reasoning. Similar trends are observed for LLaVA-1.5 13B and Qwen2-VL 7B, with details availble in Appendix C.

Table 4: **Performance comparison of strategies**: increasing visual attention only (Inc. V.), recalibrating visual attention only (Cal. V.), and increasing both visual and query attention (Ours).

| Model | Setup | MMHal-Bench | | POPE | |
|---|---|---|---|---|---|
| | | Hal. ($\downarrow$) | Score ($\uparrow$) | F1 ($\uparrow$) | Acc. ($\uparrow$) |
| LLaVA-1.5 7B | Inc. V. | 60.8 | 2.36 | 82.3 | 79.3 |
| | Cal. V. | 61.5 | 2.32 | 82.4 | 79.5 |
| | **Ours** | **57.3** | **2.48** | **83.8** | **81.9** |
| LLaVA-1.5 13B | Inc. V. | 59.2 | 2.51 | 81.3 | 77.6 |
| | Cal. V. | 59.8 | 2.46 | 81.6 | 78.0 |
| | **Ours** | **55.8** | **2.72** | **82.1** | **78.9** |
| Qwen2-VL 7B | Inc. V. | 35.2 | 3.41 | 85.3 | 86.0 |
| | Cal. V. | 31.9 | 3.56 | 85.8 | 86.4 |
| | **Ours** | **27.5** | **3.58** | **86.8** | **86.9** |

**Computation Efficiency.** We benchmark GIFT's computational overhead using LLaVA-1.5 7B on MMHal-Bench, measuring latency relative to greedy decoding with a fixed output length of 32 tokens, which approximates the average output length under greedy decoding. As shown in Figure 6, GIFT runs only at 1.13x the latency of greedy decoding, compared to 1.56x for Rel-Attn, 11.10x for VAR, and 1.01x for VAF. Despite being slightly slower than VAF, GIFT consistently outperforms all baselines across vision-hallucination datasets and models, offering a strong tradeoff between efficiency and performance.

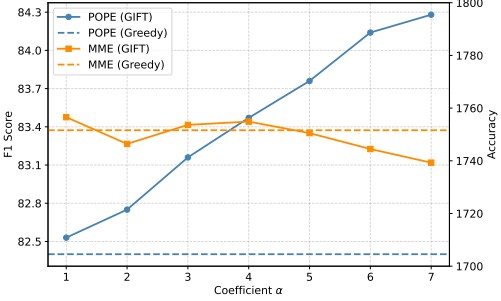

Figure 5: Performance of LLaVA-1.5 7B on the POPE and MME datasets with varying enhancement coefficients $\alpha$.

Figure 6: Relative inference latency of different methods compared to standard greedy decoding (greedy decoding = 1.0).

## 7 CONCLUSION

This work identifies critical limitations that existing inference-time hallucination mitigation methods for vision-language models (VLMs) fail to address simultaneously, including visual attention sink, low visual contribution, and imbalanced cross-modal fusion. To address these challenges, we introduce **G**aze Sh**i**ft-Guided Cross-modal **F**usion Enhancemen**t** (**GIFT**), a simple yet effective approach that constructs a holistic visual saliency map by tracking "gaze shifts" during user query processing and uses it to enhance both visual and query attentions at each decoding step. Extensive experiments show that GIFT reduces hallucination across models and datasets by up to 20.7%, while maintaining general vision-language reasoning performance with low computational overhead.

We also acknowledge a primary limitation of GIFT. The method relies heavily on the user query, and vague, ambiguous, or visually irrelevant queries may result in inaccurate visual saliency maps, reducing the effectiveness of hallucination mitigation. Future work will focus on improving the identification of vision-relevant, information-rich query tokens, potentially through a small fine-tuned auxiliary model, and on developing strategies for handling cases where visual information is not required to answer the query.

## USE OF LARGE LANGUAGE MODELS (LLMs)

In this work, we used large language models (LLMs) as a tool to polish writing. The LLM was not involved in developing research ideas, conducting experiments, or analyzing results. Its contributions were restricted to language-level assistance only.

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

## A  DATASETS

**POPE.**  The Polling-based Object Probing Evaluation (POPE) (Li et al., 2023d) assesses object hallucination in VLMs using binary questions (e.g., "Is there a frisbee in the image?"). Objects are drawn from three splits: *random*, *popular*, and *adversarial*, corresponding respectively to randomly chosen missing objects, frequently occurring objects, and co-occurring but absent objects. POPE uses images from MSCOCO (Lin et al., 2014), A-OKVQA (Schwenk et al., 2022), and GQA (Hudson & Manning, 2019), resulting in nine splits, each containing 500 MSCOCO images with six questions per image. Results are reported as macro-averaged F1 and accuracy over all splits.

**CHAIR.**  CHAIR (Captioning Hallucination Assessment with Image Relevance) (Rohrbach et al., 2018) contains 500 images for evaluating hallucination in image captioning. We use two metrics: $C_I$, which measures the proportion of hallucinated objects among all mentioned objects in captions, and $C_S$, which measures the proportion of captions containing at least one hallucinated object. Formally, these are defined as:

$$C_I = \frac{|\text{hallucinated objects}|}{|\text{all mentioned objects}|}, \quad C_S = \frac{|\text{captions with hallucinated objects}|}{|\text{all captions}|}$$

**MMHal-Bench.**  MMHal-Bench (Sun et al., 2023) is a benchmark designed to evaluate hallucination in vision language models (VLMs). It contains 96 challenging questions based on images from the OpenImages dataset (Kuznetsova et al., 2020), each paired with a corresponding ground-truth answers and annotated image content. Model responses are scored using GPT-4 through a pre-defined prompt that assesses both informativeness and hallucination.

**MME.**  MME (Fu et al., 2023) is a benchmark designed to assess both perception and cognition capabilities of vision language models across 14 subtasks. Each subtask evaluates a specific aspect of visual understanding or reasoning capability. For all experiments, We report performance using the accuracy metric as defined in the original paper.

**SEED-Bench.**  SEED-Bench (Li et al., 2023a) is a comprehensive benchmark designed to evaluates general vision-language reasoning capabilities. It contains 19,000 multiple-choice questions spanning 12 evaluation dimensions, covering both image and video modalities. In this work, we focus exclusively on the image modality and report model performance using accuracy.

## B  IMPLEMENTATION DETAILS

For all experiments, we employ greedy decoding with eager attention computation, and run inference on a single NVIDIA A100 Tensor Core GPU (40GB) instance to ensure reproducibility and fair comparisons across models and baselines. We use float16 precision for LLaVA-1.5 7B and 13B, and bfloat16 for Qwen2-VL.

For POPE, different baselines append varying suffixes to the questions, such as "Please just answer yes or no." or "Answer the question using a single word or phrase.", leading to substantial variation in evaluation results. To ensure fair comparison, we use the original dataset questions without modification. For MMHal-Bench, since the original GPT-4 version has been deprecated, we use GPT-4.1 (*gpt-4.1-2025-04-14*) for scoring. To account for the inherent randomness in GPT-4.1 scoring outputs, each evaluation is repeated five times, and the results are averaged. We set the max number of new tokens to 10 for POPE, MME, and SEED-Bench, and to 1024 for CHAIR and MMHal-Bench.

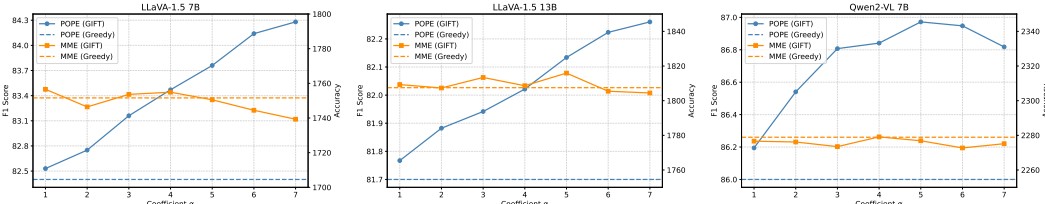

Figure 7: Performance on the POPE and MME datasets with varying enhancement coefficients $\alpha$ across models.

## C  HYPERPARAMETER TUNING

Our method involves four key hyperparameters: (1) the layer used to compute the visual saliency map, (2) the layers selected for cross-modal fusion enhancement, (3) the visual attention enhancement coefficient $\alpha$, and (4) the query attention enhancement coefficient $\beta$.

**Layer for Computing Visual Saliency Map.**   We select the saliency map computation layer as the one exhibiting the largest positive changes in visual token attention when processing information-rich user query tokens, as discussed in Section 3. Based on Figure 2, this corresponds to layer 11 for LLaVA-1.5 7B, layer 10 for LLaVA-1.5 13B, and layer 14 for Qwen2-VL 7B.

**Layers for Cross-Modal Fusion Enhancement.**   Since the benchmarks we consider lack dedicated validation sets for hyperparameter tuning, we follow Kang et al. (2025) by randomly sample 10% of the POPE and MME datasets as "pseudo-validation" sets for tuning, applying the resulting hyperparameters to all benchmark samples. Given the large number of transformer layers across models, we restrict the grid search for cross-modal fusion enhancement layers: the start layer is chosen between the first layer reaching a visual attention proportion of 0.2 and the peak layer, and the end layer is chosen between the peak layer and the last layer reaching 0.2, based on Figure 4. Following this procedure, we select layers 12-22 for LLaVA-1.5 7B, layers 14-20 for LLaVA-1.5 13B, and layers 5-18 for Qwen2-VL.

**Visual Attention Enhancement Coefficient.**   To tune the visual attention enhancement coefficient $\alpha$, we vary its value from 1.0 to 7.0 on the POPE and MME datasets, evaluating the trade-off between hallucination mitigation and reasoning performance. Results across all three models are shown in Figure 7. Based on these results, we set $\alpha = 5.0$ for LLaVA 7B and 13B, and $\alpha = 4.0$ for Qwen2-VL 7B. The higher value required for LLaVA reflects its lower original visual and query token attention compared to Qwen, as shown in Figure 4, which necessitates stronger enhancement.

**Query Attention Enhancement Coefficient.**   For query token attention enhancement, we set $\beta = 1.0$ to preserve the original cross-modal fusion balance. Investigating whether this balance is truly optimal is left for future work.

## D  LIMITATIONS AND FUTURE WORK

We acknowledge several limitations to address in future work. First, our method relies heavily on the user query; vague, ambiguous, or visually irrelevant queries may result in inaccurate visual saliency maps, reducing the effectiveness of hallucination mitigation. Future work will focus on improving the identification of vision-relevant, information-rich query tokens, potentially through a small fine-tuned auxiliary model, and on developing strategies for handling cases where visual information is not required to answer the query. Second, not all decoding steps require attention to visual inputs, as some steps primarily involve reasoning. Developing a method to dynamically determine when to enhance cross-modal fusion is left for future work. Third, while the computational overhead is relatively low, inference is still 13% slower than standard greedy decoding. This overhead can be reduced by adopting more efficient attention computation mechanisms or pruning layers during visual saliency map computation, and by partially reusing the computed key-value caches for layers preceding the start of cross-modal fusion enhancement.

# E  QUALITATIVE ANALYSIS

Figure 8-12 illustrate example outputs from the MMHal-Bench dataset, comparing our method, GIFT, with standard greedy decoding. GIFT effectively mitigates hallucination by accurately describing facts present in the image and correctly identifying absent entities. For instance, in Figure 9, the user query asks for a description of the person playing tennis, while the image only contains a tennis racket and a ball without any person. GIFT correctly responds that no person is present, whereas greedy decoding assumes a person is in the scene and hallucinates details based on visual cues and model's knowledge priors. This improvement can be attributed to GIFT's cross-modal fusion balance, which enhances both visual and query attention for more accurate and tightly integrated representations of visual and textual information. Figure 12 shows that, although GIFT generates fewer hallucinations than greedy decoding, some errors remain, highlighting the need for further refinement.

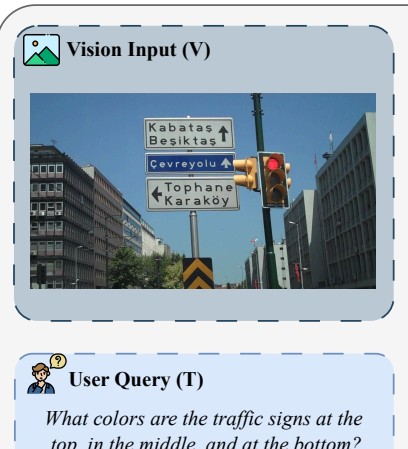

**Vision Input (V)**

**User Query (T)**

*What colors are the traffic signs at the top, in the middle, and at the bottom?*

**Ground Truth:**
The traffic signs at the top and bottom are white, while the one in the center is blue.

**Greedy Decoding:**
The traffic signs at the top, **in the middle**, and at the bottom are **white with black text**.

**GIFT (Ours):**
The top, the traffic sign is white with black text. In the middle, the traffic sign is blue with white text. At the bottom, the traffic sign is white with black text.

Figure 8: An example from the MMHal-Bench dataset. Hallucinations are highlighted in red.

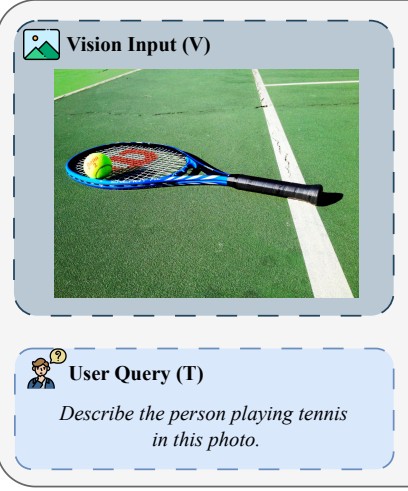

**Vision Input (V)**

**User Query (T)**

*Describe the person playing tennis in this photo.*

**Ground Truth:**
No, there is only a tennis racket and a ball on the ground. No one is playing tennis in this photo.

**Greedy Decoding:**
**The the photo, there is a person playing tennis. The person is holding** a blue and white tennis racket with a black grip. **They are standing on a green tennis court, and the ball is in the air, indicating that they are in the middle of a serve or a return. The person is wearing a white shirt and black shorts, and they appear to be focused on the game.**

**GIFT (Ours):**
There is no person in the photo.

Figure 9: An example from the MMHal-Bench dataset. Hallucinations are highlighted in red.

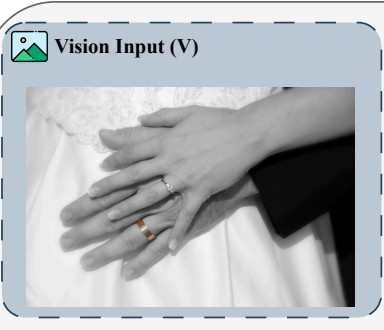

**Vision Input (V)**

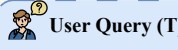
**User Query (T)**

*Describe the details in this photo.*

**Ground Truth:**
In the photo, a woman's hand is holding a man's hand. The woman's hand is wearing a diamond wedding ring, while the man's hand is wearing a simple wedding band. It seems that the woman is in a white dress and the man is in a black suit, but only a part can be seen in this photo. The image is black and white, except the wedding ring and band. This photo is likely taken in a wedding.

**Greedy Decoding:**
The image is a black-and-white photograph that captures a close-up view of two hands resting on a white surface, likely a wedding dress. The hands are positioned in a way that they overlap, **with the left hand on top of the right hand**. Both hands are adorned with rings, indicating a wedding or engagement setting. The left hand, which is on top, has a ring on the ring finger, **while the right hand, which is underneath, has a ring on the middle finger**. The background is blurred, focusing attention on the hands and the rings. The overall composition suggests a moment of celebration or commitment, likely a wedding ceremony.

**GIFT (Ours):**
The image is a black and white photograph that captures a close-up view of two hands resting on top of each other. The hands belong to a bride and groom, as indicated by the presence of wedding rings. The bride's hand is adorned with a diamond engagement ring and a wedding band, while the groom's hand features a simple wedding band. The background is a white fabric, likely the bride's wedding dress, which adds to the formal and celebratory atmosphere of the scene. The photograph focuses on the the hands and rings, emphasizing the significance of the wedding ceremony.

Figure 10: An example from the MMHal-Bench dataset. Hallucinations are highlighted in red.

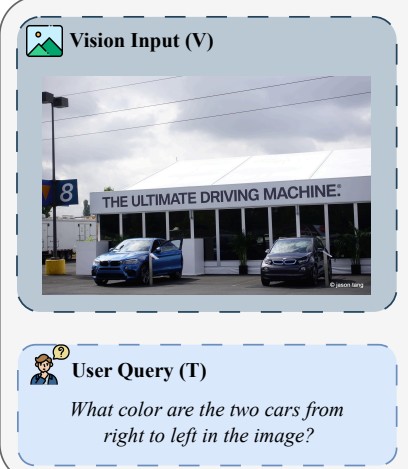

**Vision Input (V)**

**User Query (T)**

*What color are the two cars from right to left in the image?*

**Ground Truth:**
The two cars from right to left in the image are black and blue.

**Greedy Decoding:**
The two cars from right to left in the image are **black**.

**GIFT (Ours):**
The two cars from right to left in the image are black and blue.

Figure 11: An example from the MMHal-Bench dataset. Hallucinations are highlighted in red.

**Vision Input (V)**

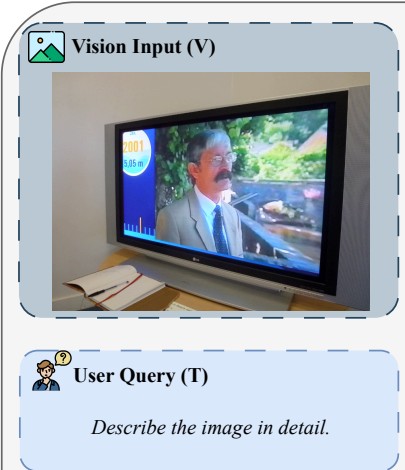

**User Query (T)**

*Describe the image in detail.*

**Ground Truth:**
In this image, I see a book, a pen, and a television on a table. The TV screen shows a man wearing a tie and a suit speaking in a park.

**Greedy Decoding:**
The image shows a large flat-screen television **mounted on a wall**. The television is displaying a scene with a man in a suit and tie, who appears to be speaking or present. The man is standing in front of a backdrop that includes greenery and a body of water, possibly a river or a lake. The screen also shows a graphical representation of water levels, with a blue bar indicating the water level at 5.05 meters in **January 2**. The television is placed on a wooden surface, possibly a desk or a table, and **there is a notebook and a pen on the surface to the left of the television**. The notebook is open, and the pen is resting on it. The overall setting suggests a casual or professional environment, possibly an office or a study room.

**GIFT (Ours):**
The image depicts a television set **mounted on a wall**, displaying a scene with a man in a suit and tie. The man appears to be speaking or present in a formal setting, **possibly a newsroom or a professional environment**. The television screen shows a graphical representation of water levels, with a blue bar indicating the water level at 5.05 meters. The background of the screen includes greenery, suggesting an outdoor or natural setting. \n\nIn the foreground, there is a desk with a notebook and a pen on it. The notebook is open, and the pen is placed on top of it. The desk appears to be made of wood, and the notebook is white with black lines, indicating it is likely a standard lined notebook. The pen is black and appears to be a standard ballpoint pen. The overall setting suggests a workspace or a study area.

Figure 12: An example from the MMHal-Bench dataset. Hallucinations are highlighted in red.

