# OpenReview forum: "Capturing Gaze Shifts for Guidance: Cross-Modal Fusion Enhancement for VLM Hallucination Mitigation"
_ICLR.cc/2026/Conference — Submitted to ICLR 2026_

### Official Review · Reviewer_37D4 · 2025-10-30

**Soundness:** 2
**Presentation:** 2
**Contribution:** 2
**Rating:** 2
**Confidence:** 4

**Summary:**

This paper addresses object hallucination, a phenomena that is prevalent in existing VLMs, which are typically caused by over-reliance on language priors. Authors propose Gaze Shift-Guided Cross-modal Fusion Enhancement (GIFT), a simple training-free method that pre-computes a visual saliency map by tracking positive changes in visual attention, which are further leveraged to amplify attention at decoding step. They test the effectiveness on multiple hallucination benchmarks and show effectiveness.

**Strengths:**

- Presentation. The overall presentation of this paper is clear.
- Clarity. The overall idea is straightforward and easy-to-follow.

**Weaknesses:**

- Comparison to Attention Modification Approaches. Note that there are a series of studies [A,B,C,D] that focuses on fixing the attention patterns to address object hallucination in this field, while authors ignores such discussions, which are suggested to include. This could involve discussions in related works, and performance comparisons.

[A] Seeing Far and Clearly: Mitigating Hallucinations in MLLMs with Attention Causal Decoding. CVPR 2025.
[B] Mitigating Object Hallucination via Concentric Causal Attention. NeurIPS 2024.
[C] Mitigating Object Hallucinations in Large Vision-Language Models with Assembly of Global and Local Attention. CVPR 2025.

- Performance on General Benchmarks. How are the performance gains on general benchmarks rather than hallucinations, for example, MMStar, or MMBench?

**Questions:**

See weakness 2.

---

> ### Author Response · Authors · 2025-11-22
>
> We sincerely appreciate your constructive feedback. We have made our best effort to address your concerns and questions, as detailed below.
>
> ### Weakness 1: Comparison to Related Work
> Thank you for highlighting these studies. Below we discuss how our work differs from each study and provide performance comparisons where feasible. We will incorporate these discussions and results into the paper.
>
> Work 1. Seeing Far and Clearly: Mitigating Hallucinations in MLLMs with Attention Causal Decoding. CVPR 2025.
>
> FarSight addresses attention sink by absorbing excessive attention from outlier tokens into a register buffer and redistributing it across remaining tokens. However, like prior attention sink studies [1, 2], it does not address the broader issue that visual features contribute relatively little during generation (as discussed in Section 2). GIFT explicitly addresses this limitation by identifying semantically salient visual information based on query content and enhancing visual contribution in cross-modal integration. We validated its importance in our ablation study (Section 6).
>
> We attempted to run their code for comparison but faced the same issue reported in their GitHub repository (https://github.com/FeilongTangmonash/FarSight/issues/3) that the model only outputs a “.”. Although the code is not available for comparison, we will still discuss this paper in the related work.
>
> Work 2. Mitigating Object Hallucination via Concentric Causal Attention. NeurIPS 2024.
>
> This approach is a positional alignment strategy that mitigates the impact of RoPE long-term decay in VLMs. Since it requires model retraining, it is not directly comparable to GIFT, which is applied at inference time. We will discuss this paper in the related work under training methods.
>
> Work 3. AGLA: Assembly of Global and Local Attention (CVPR 2025)
> AGLA masks image input based on similarity scores calculated by an additional model, then aggregates logits from both original and masked images. This approach requires an external model and doubles inference latency due to the second forward pass. In contrast, GIFT generates visual saliency maps directly from the model's internal representations and applies them to attention mechanisms, resulting in only 1.13× latency overhead.
>
> In addition, we compared GIFT against AGLA using LLaVA 1.5 models across hallucination datasets. Qwen2-VL is not supported in their code. Results below demonstrate that GIFT consistently outperforms AGLA across all benchmarks.
>
> |Model|Method|CHAIR||POPE||MMHal-Bench||
> |---|---|---|---|---|---|---|---|
> | | |  CHAIR_S ↓ | CHAIR_I ↓ | F1 ↑ | Acc. ↑ |  Hal. ↓ | Score ↑ |
> |LLaVA 1.5 7B| AGLA | 50 | 15.2 | 82.6 | 79.6 | 64.6 | 2.22 |
> | | GIFT | **39.8** | **10.6** | **83.8** | **81.9** | **57.3** | **2.48** |
> | LLaVA 1.5 13B | AGLA | 45.8 | 13.0 | 81.8 | 78.1 | 59.4 | 2.47|
> | | GIFT | **39.6**| **11.9** | **82.1** | **78.9** | **55.8** | **2.72** |
>
> We will incorporate discussions and results of these related works into our paper.
>
> [1] Kang, S., Kim, J., Kim, J., & Hwang, S. J. (2025). See what you are told: Visual attention sink in large multimodal models. arXiv preprint arXiv:2503.03321.
>
> [2] Zhu, Y., Tao, L., Dong, M., & Xu, C. (2025). Mitigating object hallucinations in large vision-language models via attention calibration. arXiv preprint arXiv:2502.01969.
>
>
> ### Weakness 2: Performance on General Benchmarks
>
> Thank you for this important question about performance on general benchmarks.
> We do not anticipate significant performance gains on general benchmarks like MMStar or MMBench, as errors in these datasets typically stem from reasoning limitations rather than perceptual hallucinations. Our primary goal is to maintain reasoning capability while reducing hallucinations.
>
> In the original submission, we have evaluated GIFT on general benchmarks including SEED-Bench and MME (Table 3, Section 5.2). GIFT consistently achieves performance comparable to greedy decoding across all datasets and models, demonstrating minimal impact on reasoning capabilities. In contrast, other baseline methods show mixed results, suggesting they face the inherent trade-off between perception and reasoning, a key challenge in hallucination mitigation.
>
> Following your suggestion, we conducted additional experiments on MMBench and MMStar using LLaVA-1.5 7B, LLaVA-1.5 13B, and Qwen2-VL 7B. The results below confirm that GIFT maintains comparable performance to the greedy baseline across all models, further validating its ability to preserve reasoning capabilities.
>
> | Model | Method |  MMBench | MMStar |
> |---|---|---|---|
> | LLaVA 1.5 7B | Greedy | 73.1 | 31.9 |
> |  | GIFT | 73.1 | 31.8 |
> | LLaVA 1.5 13B | Greedy | 75.6 | 33.5 |
> |  | GIFT |75.8 | 33.7 |
> | Qwen2 VL 7B | Greedy | 84.6 |  56.9|
> |  | GIFT | 84.6 |  57.3 |
>
> We hope this analysis could address your concern. We welcome any follow-up questions!

---

> ### Author Response · Authors · 2025-11-28
>
> Dear Reviewer 37D4,
>
> We sincerely appreciate your constructive comments. We would greatly appreciate it if you could review our responses at your earliest convenience and share any further feedback you may have. Thank you!

---

### Official Review · Reviewer_aN7r · 2025-10-31

**Soundness:** 3
**Presentation:** 3
**Contribution:** 3
**Rating:** 4
**Confidence:** 3

**Summary:**

This paper addresses the critical issue of hallucination in VLMs. The authors propose GIFT (Gaze Shift-Guided Cross-Modal Fusion Enhancement), a lightweight inference-time method inspired by human visual gaze dynamics to tackle: (1) misallocated attention to irrelevant visual regions (2) over-reliance on linguistic priors (3) imbalanced cross-modal fusion.

**Strengths:**

1.GIFT introduces a human-inspired "gaze shift" tracking approach that addresses a critical gap in existing work: static attention averaging (used by baselines like VAF) often misallocates attention to irrelevant regions.

2.It integrates into existing VLMs without retraining, unlike training-based methods that incur high computational costs.

3.It consistently improves performance across diverse VLMs (LLaVA-1.5 7B/13B, Qwen2-VL 7B) and tasks (object detection, captioning, VQA), demonstrating its versatility.

**Weaknesses:**

1.GIFT heavily relies on "information-rich query tokens" (identified via POS tagging) to compute accurate saliency maps. The authors acknowledge that vague, ambiguous, or visually irrelevant queries (e.g., "Describe this image" without specific cues) may lead to inaccurate maps and reduced hallucination mitigation. However, they do not provide concrete strategies to handle such cases—e.g., no analysis of performance on low-specificity queries or a fallback mechanism for query-scarce scenarios.

2.While the authors tune key hyper-parameters, they lack a deeper analysis of how these choices generalize.

3.GIFT is compared to three baselines (VAF, Rel-Attn, VAR) but not to recent state-of-the-art contrastive decoding methods[1,2]. These methods reduce hallucination by contrasting outputs with perturbed visual inputs and have shown strong performance on VLM hallucination tasks. Omitting this comparison limits the paper’s ability to position GIFT against the broader landscape of mitigation strategies.

[1] Mitigating Object Hallucinations in Large Vision-Language Models Through Visual Contrastive Decoding.

[2] Self-Introspective Decoding: Alleviating Hallucinations for Large Vision-Language Models.

**Questions:**

please refer to weaknesses.

---

> ### Author Response · Authors · 2025-11-22
>
> We sincerely appreciate your constructive feedback. We have made our best effort to address your concerns and questions, as detailed below.
>
> ### Weakness 1: Reliance on Information-Rich Queries
> Thank you for this important observation.
>
> First, we acknowledge that unclear writing may have caused confusion about the definition of "visually irrelevant queries". We define them as those where answering does not depend on visual information, or where substantial portions are unrelated to the image content (e.g., retrieved text documents). Such queries may produce inaccurate or unnecessary visual saliency maps, causing the model to focus on incorrect image regions or attend to visual information when unnecessary. We will clarify this definition in the revised paper.
>
> Queries like "Describe this image," while general, remain vision-dependent and fall within our method's intended scope. We have validated GIFT's effectiveness on such cases, as our method outperforms all baselines on the CHAIR dataset for image captioning tasks (Table 2), demonstrating robust performance even with general vision-dependent queries.
>
> Regarding concrete strategies for handling truly vision-irrelevant queries, we propose the following detailed approach for future work:
> We plan to enhance the identification of vision-relevant, information-rich query tokens through LLM prompting. To address latency concerns, we will explore fine-tuning a lightweight auxiliary model for token classification, where each token's score weights its contribution to saliency map computation. When no query tokens relate to the image content, we will bypass visual saliency map computation entirely, preventing unnecessary computational overhead and potential error propagation.
>
> However, to our knowledge, no existing benchmark specifically targets scenarios where answering does not depend on visual information despite image input being provided, or where substantial query portions are unrelated to image content. Creating such a dataset represents valuable future work that would benefit the broader research community.
>
> We sincerely appreciate this feedback and will incorporate the concrete strategy into the paper.

---

> ### Author Response · Authors · 2025-11-22
>
> ### Weakness 2: Generalizability of Hyper-parameter Tuning
> Thank you for this important question regarding hyper-parameter generalization.
>
> While unified hyper-parameters cannot apply across all models due to architectural differences (e.g., varying layer counts and training processes), our tuning methodology generalizes effectively and remains cost-efficient. We have documented the complete tuning procedure in Appendix C and will expand these details in the revision.
>
> Our method involves three key hyper-parameters, each demonstrating low sensitivity and strong generalization:
> 1. Saliency Map Extraction Layer
> We identify the optimal layer where visual attention undergoes the most dynamic realignment during query processing, measured by the positive changes in attention allocated to the visual tokens. Using 50 examples from TextVQA's training set for measurement (as detailed in Section 3), this method consistently works across different model architectures, demonstrating robust generalization without requiring extensive tuning.
>
> 2. Enhancement Coefficient \alpha
> We tune \alpha from 1.0 to 7.0 (step size 1.0) using 10% of POPE and MME datasets to balance hallucination reduction with reasoning capability. Performance curves across models (LLaVA-1.5 7B in Figure 5; LLaVA-1.5 13B and Qwen2-VL in Appendix C) exhibit consistent trends without sensitive fluctuations, confirming robustness. As \alpha increases, hallucination rates steadily decrease. However, only when \alpha exceeds 5.0, MME performance drops below greedy decoding due to excessive visual emphasis that compromises reasoning, indicating a clear range that generalizes well across models.
>
> 3. Cross-Modal Enhancement Layers
> We apply the same tuning sets as (2). To address the large search space, we constrain the grid search: start layers are selected between the first layer reaching 0.2 visual attention proportion and the peak layer; end layers between the peak layer and the last layer reaching 0.2. 0.2 is used in VAR [1] to select image-centric attention heads.
>
> Using LLaVA-1.5 7B, we further conducted robustness testing by progressively expanding or narrowing the selected layer boundary (layer 12-22) and evaluating on both POPE and SEED-Bench. Results below demonstrate that hallucination performance improves steadily and saturates at layers 13-21, while reasoning capability remains stable until layers 12-22. Additionally, shifting the boundary by 1-2 layers does not significantly affect performance. This analysis confirms low hyper-parameter sensitivity and shows that the method is robust to boundary variations.
>
> | Enhancement Layers | POPE | SEED-Bench |
> |---|---|---|
> | None | 82.4 | 65.5 |
> | 17 | 82.3 | 65.6 |
> | 16-18 | 83.0 | 65.7 |
> | 15-19 | 83.5 | 65.6 |
> | 14-20 | 83.2 | 65.6 |
> | 13-21 | 83.7 | 65.4 |
> | 12-22 (selected) | 83.8 | 65.6 |
> | 11-23 | 84.0 | 65.2 |
> | 10-24 | 83.6 | 65.2 |
> | 9-25 | 84.0 | 65.0 |
> | 8-26 | 83.9 | 65.1 |
> | 7-27 | 83.9 | 65.0 |
> | 6-28 | 83.8 | 64.6 |
> | 5-29 | 83.8 | 64.4 |
> | 4-30 | 84.0 | 63.9 |
>
> To further validate the generalizability of our hyperparameter tuning method, we applied the same hyperparameters from the paper to two additional benchmarks: the hallucination benchmark AMBER [2], which covers both discriminative and generative tasks, and the general benchmark MMBench [3]. We also applied the tuning methodology detailed in Appendix C to evaluate Qwen3-VL 8B, a new model released one month ago with enhanced capabilities. Results below demonstrate that our tuning method and hyper-parameter choices generalize well across models and datasets.
>
> **Hallucination Benchmark**
> | Model | Method | AMBER |  |  |
> |---|---|---|---|---|
> | |  | AMBER ↑  | Dis. ↑ | Gen. ↓ |
> | LLaVA 1.5 7B | Greedy | 83.9 | 74.9 | 7.2 |
> |  | GIFT | **87.2** | **80.9** | **6.6** |
> | LLaVA 1.5 13B | Greedy | 83.4 | 73.5 | 6.7 |
> |  | GIFT | **85.7** | **76.8** | **5.5** |
> | Qwen2 VL 7B | Greedy | 84.1 | 74.4 | 6.2 |
> |  | GIFT | **85.4** | **76.2** | **5.5** |
> | Qwen 3 VL 8B | Greedy | 80.2 | 68.3 | 7.9 |
> |  | GIFT | **83.6** | **74** | **6.8** |
>
> **General Benchmark**
> | Model | Method  | MMBench |
> |---|---|---|
> | LLaVA 1.5 7B | Greedy | 73.1 |
> |  | GIFT | 73.1 |
> | LLaVA 1.5 13B | Greedy | 75.6 |
> |  | GIFT | 75.8 |
> | Qwen2 VL 7B | Greedy | 84.6 |
> |  | GIFT | 84.6 |
> | Qwen 3 VL 8B | Greedy | 86.9 |
> |  | GIFT | 87.2 |
>
> In summary, our systematic tuning approach ensures robust performance across architectures while maintaining computational efficiency and minimal sensitivity to hyperparameter choices.
>
> [1] Kang, Seil, et al. "See what you are told: Visual attention sink in large multimodal models." arXiv preprint arXiv:2503.03321 (2025).
>
> [2] Wang, Junyang, et al. "An llm-free multi-dimensional benchmark for mllms hallucination evaluation." CoRR (2023).
>
> [3] Liu, Yuan, et al. "Mmbench: Is your multi-modal model an all-around player?." European conference on computer vision. Cham: Springer Nature Switzerland, 2024.

---

> ### Author Response · Authors · 2025-11-22
>
> ### Weakness 3: Comparison to Contrastive Decoding Approaches
>
> Thank you for this valuable suggestion. We agree that including comparisons with contrastive decoding methods would better position GIFT within the broader mitigation landscape.
>
> We initially focused our comparisons on attention-based methods (VAF, Rel-Attn, VAR) as they represent the most directly comparable approaches to GIFT. We mentioned the contrastive decoding methods in our related work but did not include them as primary baselines for two reasons: (1) they incur significant computational overhead due to generating counterpart output logits, and (2) they operate through different mechanisms (contrastive decoding vs. attention modification).
>
> Following your recommendation, we conducted comprehensive evaluations comparing GIFT with both Visual Contrastive Decoding and Self-Introspective Decoding to provide a more complete performance landscape. Since their released code does not support Qwen2-VL, we evaluated these methods on LLaVA-1.5 7B. Results below show that GIFT consistently achieves better performance across hallucination datasets.
>
> **Hallucination Benchmarks**
> |  Method | CHAIR |  | POPE |  | MMHal-Bench  |  |
> |---|---|---|---|---|---|---|
> |  | CHAIR_S ↓ | CHAIR_I ↓ | F1 ↑ | Acc. ↑ |  Hal. ↓ | Score ↑ |
> | VCD | 52.2 | 16.3 | 80.9 | 77.7 | 60.5 | 2.37 |
> |SID | 49.2 | 13.9 | 83.0 | 80.7 | 60.5 | 2.42 |
> |GIFT | **39.8** | **10.6** | **83.8** | **81.9** | **57.3** | **2.48**|
>
> We will incorporate these comparisons into the paper to provide readers with a comprehensive view of GIFT's performance relative to the broader landscape.
>
> We hope these analyses could address your concerns. We welcome any follow-up questions!

---

> ### Author Response · Authors · 2025-11-28
>
> Dear Reviewer aN7r,
>
> We sincerely appreciate your constructive comments. In response, we have added concrete strategies for handling vague, ambiguous, or visually irrelevant queries, added hyper-parameter tuning analysis to validate its generalizability and robustness, and added comparison results against contrastive decoding methods (VCD, SID)
>
> We would greatly appreciate it if you could review our responses at your earliest convenience and share any further feedback you may have. Thank you!

---

### Official Review · Reviewer_nbc4 · 2025-10-31

**Soundness:** 3
**Presentation:** 3
**Contribution:** 2
**Rating:** 4
**Confidence:** 4

**Summary:**

This paper proposes GIFT, an inference-time hallucination mitigation method for VLMs. The key novelty lies in creating a visual saliency map by tracking positive changes in visual attention during comprehension of information-rich query tokens. Unlike previous approaches that only enhance visual attention, GIFT also proportionally adjusts query token attention to preserve cross-modal fusion balance, reducing the risk of visual attention sink and low visual contribution. Evaluations on multiple hallucination benchmarks (CHAIR, POPE, MMHal-Bench) and general VLM benchmarks (MME, SEED-Bench) show significant decrease in hallucination rates with minimal impact on overall reasoning capabilities. The method is lightweight, training-free, and generalizes across different VLM architectures.

**Strengths:**

1. Clear and intuitive idea: The gaze shift concept is easy to understand and well-motivated by human visual attention behavior.

2. Addresses multiple issues simultaneously: Tackles visual attention sink, low visual contribution, and imbalanced cross-modal fusion, which existing methods often address in isolation.

3. Low computational overhead: Achieves improvements with modest runtime increase compared to greedy decoding.

4. Comprehensive evaluation: Experiments span multiple benchmarks, models, and hallucination types, with ablation to validate design choices.

**Weaknesses:**

1. Experimental setting is somewhat outdated: The chosen base VLMs (LLaVA-1.5 series and Qwen2-VL) were released over a year ago. More recent models—such as LLaVA-Next, InternVL—implement Dynamic High Resolution image processing, which could impact saliency computation. Testing the method on these architectures would strengthen claims about generality.

2. Limited hallucination benchmarks: Evaluation could include newer datasets such as HallusionBench or other recent challenging hallucination tasks to better measure robustness.

3. Interpretability validation missing: Since the method relies heavily on saliency maps, adding segmentation-based experiments from classic interpretability literature could reveal whether human-perceived semantic enhancement indeed contributes to hallucination reduction[1, 2].

4. Hyperparameters vary per model without explicit robustness test: While tuning is explained, an ablation on sensitivity to hyperparameter changes would reinforce the robustness claim.

[1] Optimizing Relevance Maps of Vision Transformers Improves Robustness

[2] Generic Attention-model Explainability for Interpreting Bi-Modal and Encoder-Decoder Transformers

**Questions:**

See weaknesses.

---

> ### Author Response · Authors · 2025-11-22
>
> We sincerely appreciate your valuable feedback. We have made our best effort to address your concerns and questions, as detailed below.
>
> ### Weakness 1: Experimentation with More Advanced Models
> Thank you for the thoughtful suggestion to evaluate our method on more recent VLM architectures. In response, we expanded our experiments to include two new models:
> 1. Qwen3-VL 8B, a model introduced just one month ago with significantly enhanced capabilities.
> 2. InternVL3 8B, which incorporates dynamic high-resolution processing, a promising direction for reducing hallucination by providing clearer and more detailed visual inputs. This is conceptually related to one of our baselines, Rel-Attn [1], which crops small, informative image patches as additional inputs. This model creates an interesting setting for testing whether our method can offer additional gains.
>
> Our results below show that GIFT consistently improves performance on Qwen3-VL 8B across all hallucination benchmarks. For InternVL3 8B, we observe solid improvements on MMHal-Bench and CHAIR, with more modest gains on AMBER and POPE. We believe this is likely because the dynamic high-resolution mechanism, which splits the image into multiple detailed patches as additional inputs, already substantially mitigates hallucination, especially on object detection datasets like POPE. Interestingly, under greedy decoding, InternVL3 8B achieves stronger hallucination performance but weaker general-purpose performance compared to Qwen3-VL 8B.
>
> **Hallucination Benchmarks**
> | Model | Method | AMBER |  |  | CHAIR |  | POPE |  | MMHal-Bench  |  |
> |---|---|---|---|---|---|---|---|---|---|---|
> | |  | AMBER ↑  | Dis. ↑ | Gen. ↓ | CHAIR_S ↓ | CHAIR_I ↓ | F1 ↑ | Acc. ↑ |  Hal. ↓ | Score ↑ |
> | LLaVA 1.5 7B | Greedy | 83.9 | 74.9 | 7.2 | 50.2 | 15.4 | 82.4 | 79.5 | 65.2 | 2.22 |
> |  | GIFT | **87.2** | **80.9** | **6.6** | **39.8** | **10.6** | **83.8** | **81.9** | **57.3** | **2.48** |
> | LLaVA 1.5 13B | Greedy | 83.4 | 73.5 | 6.7 | 46.8 | 13.1 | 81.7 | 78.2 | 56.2 | 2.61 |
> |  | GIFT | **85.7** | **76.8** | **5.5** | **39.6** | **10.9** | **82.1** | **78.9** | **55.8** | **2.72** |
> | Qwen2 VL 7B | Greedy | 84.1 | 74.4 | 6.2 | 24.8 | 9.1 | 86 | 86.5 | 32.7 | 3.53 |
> |  | GIFT | **85.4** | **76.2** | **5.5** | **21.2** | **7.7** | **86.8** | **86.9** | **27.5** | **3.58** |
> | Qwen 3 VL 8B | Greedy | 80.2 | 68.3 | 7.9 | 51.4 | 10.6 | 88.9 | 88.5 | 28.3 | 4.80 |
> |  | GIFT | **83.6** | **74** | **6.8** | **49.4** | **9.3** | **89.1** | **88.7** | **26.4** | **4.84** |
> | InternVL 3 8B | Greedy | 85.1 | 78.4 | 8.3 | 29 | 8.6 | 89.3 | 89.0 | 26.7 | 4.07 |
> |  | GIFT | **85.2** | **78.6** | **8.2** | **27.4** | **7.3** | **89.4** | **89.0** | **25.4** | **4.20** |
>
>
> Even in this setting, GIFT still provides improvements on multiple datasets, demonstrating its robustness and effectiveness. Furthermore, across both models, GIFT achieves comparable performance to greedy decoding on general reasoning benchmarks, indicating that our approach has no impact on reasoning capabilities.
>
> **General Benchmarks**
> | Model | Method | MME | SEED-Bench |
> |---|---|---|---|
> | Qwen 3 VL 8B | Greedy |  2403.5 | 78.4 |
> |  | GIFT | 2418.5 | 78.3 |
> | InternVL3 8B | Greedy |  2310.8 | 77.2 |
> |  | GIFT | 2317.7 | 77.2 |
>
> We appreciate the reviewer’s suggestion. We will incorporate these results into the paper.
>
>
> ### Weakness 2: Limited Hallucination Benchmarks
> Thank you for this valuable suggestion regarding additional hallucination benchmarks.
>
> We have expanded our evaluation to include AMBER [2], which assesses both generative and discriminative tasks, providing broader benchmark coverage. As shown in the first table under Weakness 1, our evaluation on AMBER demonstrates that GIFT consistently outperforms the greedy decoding baseline across all tested models, further confirming the robustness of our approach.
>
> Our original evaluation used three datasets that align with those used in the baseline VAR [3], covering diverse tasks including object detection, image captioning, and visual question answering. Among these datasets, MMHalBench was released contemporaneously with HallusionBench and is specifically designed to capture various question categories and topics relevant to hallucination detection.
>
> We believe these datasets collectively provide a comprehensive evaluation framework for hallucination benchmarks. We did not include HallusionBench because it is designed to evaluate both image and video modalities, whereas our method focuses exclusively on image modality.
>
> [1] Zhang, Jiarui, et al. "Mllms know where to look: Training-free perception of small visual details with multimodal llms." arXiv preprint arXiv:2502.17422 (2025).
>
> [2] Wang, Junyang, et al. "An llm-free multi-dimensional benchmark for mllms hallucination evaluation." CoRR (2023).
>
> [3] Kang, Seil, et al. "See what you are told: Visual attention sink in large multimodal models." arXiv preprint arXiv:2503.03321 (2025).

---

> ### Author Response · Authors · 2025-11-22
>
> ### Weakness 3: Missing Interpretability Validation
>
> Thank you for this valuable suggestion. We agree that interpretability validation is crucial, particularly given our reliance on saliency maps.
>
> Following your recommendation, we conducted a segmentation-based evaluation using the same 1,000 examples from the MSCOCO 2014 training set as in Table 1. The evaluation in table 1 partially provides interpretability by measuring the proportion of saliency falling within ground-truth bounding boxes, normalized by each box's relative area. We used standard object detection queries such as "Is there a {object} in the image?"
>
> We compared two approaches: the vanilla baseline, which averages visual token attention across all query tokens, and our proposed method (GIFT), which averages positive changes in visual token attention over information-rich query tokens. Since we lack a tuning set for threshold selection, we primarily report Average Precision. GIFT achieves **35.8 AP**, significantly outperforming the vanilla baseline's **22.6 AP**. This demonstrates that our mechanism generates saliency maps that are more semantically aligned with human-perceived object regions, confirming that improved semantic relevance actively contributes to hallucination mitigation.
>
> We acknowledge that the absolute AP values are modest. We believe this reflects the fact that the language model is not explicitly trained for segmentation. A robust saliency map for object detection should capture not only key attributes within the object but also contextual information that aids in identification, the latter of which may extend beyond bounding boxes.
>
> We will cite both suggested papers and include these new results and discussion in the final revision.

---

> ### Author Response · Authors · 2025-11-22
>
> ### Weakness 4: Sensitivity of Hyper-parameter Tuning
> We appreciate this important question regarding hyper-parameter sensitivity.
>
> Our tuning methodology generalizes effectively across models, remains cost-efficient, and exhibits low sensitivity. We have documented the complete tuning procedure in Appendix C and will expand these details with additional sensitivity analysis below in the revision.
>
> Our method involves three key hyper-parameters, each demonstrating low sensitivity and strong generalization:
> 1. Saliency Map Extraction Layer
> We identify the optimal layer where visual attention undergoes the most dynamic realignment during query processing, measured by the positive changes in attention allocated to the visual tokens. Using 50 examples from TextVQA's training set for measurement (as detailed in Section 3), this method consistently works across different model architectures, demonstrating robust generalization without requiring extensive tuning based on dataset performance.
>
> 2. Enhancement Coefficient \alpha
> We tune \alpha from 1.0 to 7.0 (step size 1.0) using 10% of POPE and MME datasets to balance hallucination reduction with reasoning capability. Performance curves across models (LLaVA-1.5 7B in Figure 5; LLaVA-1.5 13B and Qwen2-VL in Appendix C) exhibit consistent trends without sensitive fluctuations, confirming robustness. As \alpha increases, hallucination rates steadily decrease. However, only when \alpha exceeds 5.0, MME performance drops below greedy decoding due to excessive visual emphasis that compromises reasoning, indicating a clear range that generalizes well across models.
>
> 3. Cross-Modal Enhancement Layers
> We apply the same tuning sets as (2). To address the large search space, we constrain the grid search: start layers are selected between the first layer reaching 0.2 visual attention proportion and the peak layer; end layers between the peak layer and the last layer reaching 0.2. 0.2 is used in VAR [1] to select image-centric attention heads.
>
> Using LLaVA-1.5 7B, we further conducted robustness testing by progressively expanding or narrowing the selected layer boundary (layer 12-22) and evaluating on both POPE and SEED-Bench. Results below demonstrate that hallucination performance improves steadily and saturates at layers 13-21, while reasoning capability remains stable until layers 12-22. Additionally, shifting the boundary by 1-2 layers does not significantly affect performance. This analysis confirms low hyper-parameter sensitivity and shows that the method is robust to boundary variations.
>
> | Enhancement Layers | POPE | SEED-Bench |
> |---|---|---|
> | None | 82.4 | 65.5 |
> | 17 | 82.3 | 65.6 |
> | 16-18 | 83.0 | 65.7 |
> | 15-19 | 83.5 | 65.6 |
> | 14-20 | 83.2 | 65.6 |
> | 13-21 | 83.7 | 65.4 |
> | 12-22 (selected) | 83.8 | 65.6 |
> | 11-23 | 84.0 | 65.2 |
> | 10-24 | 83.6 | 65.2 |
> | 9-25 | 84.0 | 65.0 |
> | 8-26 | 83.9 | 65.1 |
> | 7-27 | 83.9 | 65.0 |
> | 6-28 | 83.8 | 64.6 |
> | 5-29 | 83.8 | 64.4 |
> | 4-30 | 84.0 | 63.9 |
>
> To further validate the low sensitivity, we applied the same hyperparameters from the paper to one more hallucination benchmark AMBER [2], which covers both discriminative and generative tasks. Results below demonstrate that our hyper-parameter choices generalize well across models and datasets.
>
> **Hallucination Benchmark**
> | Model | Method | AMBER |  |  |
> |---|---|---|---|---|
> | |  | AMBER ↑  | Dis. ↑ | Gen. ↓ |
> | LLaVA 1.5 7B | Greedy | 83.9 | 74.9 | 7.2 |
> |  | GIFT | **87.2** | **80.9** | **6.6** |
> | LLaVA 1.5 13B | Greedy | 83.4 | 73.5 | 6.7 |
> |  | GIFT | **85.7** | **76.8** | **5.5** |
> | Qwen2 VL 7B | Greedy | 84.1 | 74.4 | 6.2 |
> |  | GIFT | **85.4** | **76.2** | **5.5** |
> | Qwen 3 VL 8B | Greedy | 80.2 | 68.3 | 7.9 |
> |  | GIFT | **83.6** | **74** | **6.8** |
>
> [1] Kang, Seil, et al. "See what you are told: Visual attention sink in large multimodal models." arXiv preprint arXiv:2503.03321 (2025).
>
> [2] Wang, Junyang, et al. "An llm-free multi-dimensional benchmark for mllms hallucination evaluation." CoRR (2023).
>
>
> We hope these analyses could address your concerns. We welcome any follow-up questions. Thank you!

---

> ### Author Response · Authors · 2025-11-28
>
> Dear Reviewer nbc4,
>
> We sincerely appreciate your constructive comments. In response, we have added evaluation results for two new recent models (Qwen3-VL 8B and InternVL3 8B) to validate generalizability across models, added a new hallucination dataset to further validate generalizability across datasets, added segmentation-based evaluation to validate the contribution of human-perceived semantic enhancement, and added hyper-parameter tuning analysis to validate its robustness.
>
> We would greatly appreciate it if you could review our responses at your earliest convenience and share any further feedback you may have. Thank you!

---

### Official Review · Reviewer_iBdX · 2025-11-01

**Soundness:** 3
**Presentation:** 3
**Contribution:** 3
**Rating:** 6
**Confidence:** 4

**Summary:**

This paper introduces Gaze Shift-Guided Cross-modal Fusion Enhancement, a novel method for mitigating hallucinations in VLMs. The proposed method tracks changes in visual attention, referred to as gaze shifts, during the processing of information-rich query tokens. These gaze shifts are then used to create a visual saliency map that guides cross-modal fusion, enhancing both visual and query attention during decoding. The paper demonstrates that GIFT effectively reduces hallucination in VLMs across various tasks and datasets, providing significant improvements in hallucination mitigation while maintaining general performance with low computational overhead.

**Strengths:**

1. The idea of using gaze shifts to dynamically adjust visual attention in VLMs is a novel and promising approach. It effectively addresses key challenges in cross-modal fusion and visual attention misallocation (visual attention sink), which are critical issues in VLM performance.

2. The paper provides extensive experiments that show GIFT achieves up to 20.7% improvement in hallucination mitigation, outperforming existing methods across several vision-language datasets and models of varying architectures.

3. GIFT demonstrates that it can improve hallucination mitigation without introducing significant computational overhead, making it a practical solution for inference-time interventions in VLMs.

**Weaknesses:**

1. Some formulas are missing concluding punctuation (e.g., periods at the end of equations). Sections 5 and 6 could be merged. Both sections discuss experimental results and analyses, and their separation feels redundant. Combining them into a single cohesive section would improve the flow and clarity of the paper.

2. The experiments in the paper are mainly focused on the LLaVA model, which limits the generalizability of the results. Although the authors show promising results for LLaVA, there is no comprehensive evaluation on other popular VLMs or tasks. This raises concerns about the method's applicability to a wider range of models and real-world scenarios. The lack of a broader experimental comparison is the primary reason I am rating this paper 6/10 instead of 8/10, as it makes it difficult to assess whether GIFT is a universally applicable solution or if it is specific to certain architectures.

3. While the paper emphasizes that GIFT maintains a low computational cost compared to some baselines, it still incurs a slight increase in latency (1.13x compared to greedy decoding). However, this is not a major issue.

**Questions:**

See Weakness.

---

> ### Author Response · Authors · 2025-11-22
>
> We appreciate your constructive feedback. We have made our best effort to address your concerns and questions, as detailed below.
>
> ### Weakness 1: Formatting and Structural Suggestions
> We appreciate these valuable suggestions. We will address the issues and combine Sections 5 and 6 into a unified experimental section in the final version.
>
> ### Weakness 2: Generalizability Concern
> Thank you for raising this important concern about the generalizability of our results.
>
> First, we would like to clarify that our existing evaluation extends beyond LLaVA models. We evaluated three models: Qwen2-VL 7B, LLaVA 1.5 7B,  and LLaVA 1.5 13B, across three diverse hallucination datasets: POPE (object detection), CHAIR (image captioning), and MMHalBench (vision question answering).
>
> To further address your concerns, we conducted additional experiments on Qwen3-VL 8B, which was released 1 month ago with enhanced capabilities, and expanded our evaluation to include AMBER [1], which covers both discriminative and generative tasks, as well as MMBench [2] as a general benchmark. The results below consistently show that GIFT outperforms Greedy Decoding on hallucination datasets across all models while maintaining comparable performance on general benchmarks, indicating minimal impact on reasoning capabilities.
>
> **Hallucination Benchmarks**
> | Model | Method | AMBER |  |  | CHAIR |  | POPE |  | MMHal-Bench  |  |
> |---|---|---|---|---|---|---|---|---|---|---|
> | |  | AMBER ↑  | Dis. ↑ | Gen. ↓ | CHAIR_S ↓ | CHAIR_I ↓ | F1 ↑ | Acc. ↑ |  Hal. ↓ | Score ↑ |
> | LLaVA 1.5 7B | Greedy | 83.9 | 74.9 | 7.2 | 50.2 | 15.4 | 82.4 | 79.5 | 65.2 | 2.22 |
> |  | GIFT | **87.2** | **80.9** | **6.6** | **39.8** | **10.6** | **83.8** | **81.9** | **57.3** | **2.48** |
> | LLaVA 1.5 13B | Greedy | 83.4 | 73.5 | 6.7 | 46.8 | 13.1 | 81.7 | 78.2 | 56.2 | 2.61 |
> |  | GIFT | **85.7** | **76.8** | **5.5** | **39.6** | **10.9** | **82.1** | **78.9** | **55.8** | **2.72** |
> | Qwen2 VL 7B | Greedy | 84.1 | 74.4 | 6.2 | 24.8 | 9.1 | 86 | 86.5 | 32.7 | 3.53 |
> |  | GIFT | **85.4** | **76.2** | **5.5** | **21.2** | **7.7** | **86.8** | **86.9** | **27.5** | **3.58** |
> | Qwen 3 VL 8B | Greedy | 80.2 | 68.3 | 7.9 | 51.4 | 10.6 | 88.9 | 88.5 | 28.3 | 4.80 |
> |  | GIFT | **83.6** | **74** | **6.8** | **49.4** | **9.3** | **89.1** | **88.7** | **26.4** | **4.84** |
>
> **General Benchmarks**
> | Model | Method | MME | SEED-Bench | MMBench |
> |---|---|---|---|---|
> | LLaVA 1.5 7B | Greedy | 1751.6 | 65.5 | 73.1 |
> |  | GIFT | 1750.5 | 65.6 | 73.1 |
> | LLaVA 1.5 13B | Greedy | 1807.5 | 67.8 | 75.6 |
> |  | GIFT | 1815.9 | 67.7 | 75.8 |
> | Qwen2 VL 7B | Greedy | 2278.9 | 76.0 | 84.6 |
> |  | GIFT | 2279.1 | 76.0 | 84.6 |
> | Qwen 3 VL 8B | Greedy |  2403.5 | 78.4 | 86.9 |
> |  | GIFT | 2418.5 | 78.3 | 87.2 |
>
>
> Importantly, we applied the same hyper-parameter tuning methodology (detailed in Appendix C) to Qwen3-VL 8B, and for other models, when evaluating on AMBER, we reused the hyperparameters from our original experiments. This demonstrates the robustness of our approach and suggests that GIFT can be effectively applied to new models and datasets without extensive re-tuning.
>
> We will incorporate these results into the paper. We hope these results could address your concern and strengthen our claim that GIFT is a broadly applicable solution for reducing hallucinations in vision-language models, not limited to specific architectures or tasks. We appreciate your constructive feedback and welcome any follow-up questions.
>
> [1] Wang, Junyang, et al. "An llm-free multi-dimensional benchmark for mllms hallucination evaluation." CoRR (2023).
>
> [2] Liu, Yuan, et al. "Mmbench: Is your multi-modal model an all-around player?." European conference on computer vision. Cham: Springer Nature Switzerland, 2024.
>
> ### Weakness 3: Slight Increase in Latency
> Thank you for acknowledging that the computational cost remains reasonable. We believe it can be further reduced through several optimization strategies documented in Appendix D. For instance, during visual saliency map computation, we can adopt more efficient attention computation mechanisms or prune middle layers. Additionally, we can reuse the computed key-value caches for layers preceding the cross-modal fusion enhancement. These optimizations would make GIFT more practical for deployment while maintaining its effectiveness in reducing hallucinations.
>
> Thank you so much for your constructive feedback and we welcome any follow-up questions.

---

> ### Author Response · Authors · 2025-11-28
>
> Dear Reviewer iBdX,
>
> We sincerely appreciate your constructive comments. We would greatly appreciate it if you could review our responses at your earliest convenience and share any further feedback you may have.
>
> Thank you!

---

### Author Response · Authors · 2025-12-03
**Rebuttal Summary and Key Updates - Part 1**

Dear Area Chair,

We sincerely appreciate your efforts in this re-assignment process and understand the workload involved. To assist in your review, we summarize the key points below.

Based on reviewer feedback, the proposed method GIFT is clearly presented, well motivated, and novel, consistently outperforming baselines across all models and datasets. Regarding weaknesses, three common concerns were raised by multiple reviewers: model/dataset expansion beyond the existing 3 models and 3 hallucination datasets, hyper-parameter tuning robustness analysis, and additional related work for broader context. Individual reviewers also raised specific points based on their interests.

To address these concerns, we conducted additional experiments and believe we have addressed all weaknesses raised by the reviewers, as detailed below.

---

### Common Concerns

**Concern 1. Model and Dataset Expansion**

**Concern**: Reviewer iBdX noted that experiments primarily focus on the LLaVA model, and this limited scope is the main reason for their 6/10 rating instead of 8/10. Reviewer nbc4 requested experiments with more recent models (e.g.,
InternVL with dynamic high-resolution image processing) and newer datasets like HallusionBench to better measure robustness.

**Response**: We first clarified that our paper includes experiments with LLaVA 1.5 7B, LLaVA 1.5 13B, and Qwen2-VL 7B to measure method generalization, evaluated across three diverse hallucination datasets: POPE (object detection), CHAIR (image captioning), and MMHalBench (vision QA). MMHalBench was released contemporaneously with HallusionBench and is specifically designed to capture various question categories relevant to hallucination detection.

Additionally, as requested, we have expanded our experiments to include two new models (Qwen3-VL 8B and InternVL3 8B) and a new dataset, AMBER, which assesses both generative and discriminative tasks and was released contemporaneously with HallusionBench. We did not include HallusionBench because it evaluates both image and video modalities, whereas our method focuses exclusively on images.

Results show that GIFT consistently outperforms Greedy Decoding on hallucination datasets across all models while maintaining comparable performance on general benchmarks, demonstrating minimal impact on reasoning capabilities and
addressing concerns about method robustness.

**Concern 2. Hyperparameter Tuning Robustness and Sensitivity**

**Concern**: Reviewer nbc4 requested an ablation study on sensitivity to hyperparameter changes, and reviewer aN7r requested analysis of how the tuning process generalizes.

**Response**: We documented the complete tuning procedure in Appendix C. To demonstrate robustness, we applied the same tuning method to additional models like Qwen3-VL 8B, which also shows reduced hallucination without compromising
reasoning capability compared to greedy decoding.

To demonstrate that hyperparameter choices are not sensitive, we: (1) applied the same hyperparameters tuned for existing models to an additional hallucination dataset (AMBER) and general benchmark (MMBench), showing that all three
models (LLaVA 1.5 7B, LLaVA 1.5 13B, Qwen2-VL 7B) reduce hallucination without hurting reasoning capability; and (2) for each hyperparameter, either referenced existing robustness analysis in the paper (saliency map extraction layer and
enhancement coefficient \alpha) or added new analysis proving that moderate changes do not significantly affect results.

**Concern 3. Additional Related Work for Broader Context**

**Concern**: Reviewer aN7r suggested comparing with contrastive decoding approaches VCD and SID. Reviewer 37D4 shared three attention modification-based hallucination mitigation works [A,B,C], suggesting inclusion in either related work or
performance comparison.

[A] Seeing Far and Clearly: Mitigating Hallucinations in MLLMs with Attention Causal Decoding.

[B] Mitigating Object Hallucination via Concentric Causal Attention.

[C] Mitigating Object Hallucinations in Large Vision-Language Models with Assembly of Global and Local Attention.


**Response**: For contrastive decoding approaches, we mentioned them in related work but initially excluded them due to significant computational overhead and different mechanisms (contrastive decoding vs. attention modification). However,
we agree that comparison provides broader context. Results show that GIFT outperforms VCD and SID across hallucination datasets. We will include this comparison and expand the related work discussion.

For the three works shared by reviewer 37D4: Work A could not be evaluated due to a known GitHub issue reported in their repo. Work B is a positional alignment strategy requiring model retraining, making it not directly comparable to GIFT For Work C, our comparison shows that GIFT consistently outperforms it across datasets and models. We will incorporate these works into related work and
performance comparison accordingly.

---

### Author Response · Authors · 2025-12-03
**Rebuttal Summary and Key Updates - Part 2**

**Individual Concerns**

**Reviewer iBdX**: We will address the suggestions regarding missing concluding punctuations in equations and combining sections 5 and 6 in the final version. Regarding the slight latency increase (1.13x compared to greedy decoding), which the reviewer does not consider as a major issue, we refer to several optimization strategies documented in Appendix D that would make GIFT more practical for deployment while maintaining effectiveness.

**Reviewer nbc4**: To evaluate whether human-perceived semantic enhancement contributes to hallucination reduction, we compared two approaches on segmentation tasks: the vanilla baseline (averaging visual token attention across all query
tokens) and GIFT (averaging positive changes in visual token attention over information-rich query tokens). GIFT achieves better segmentation metrics, indicating that our mechanism generates saliency maps more semantically aligned with human-perceived object regions, confirming that improved semantic relevance actively contributes to hallucination mitigation.

**Reviewer aN7r**: The reviewer mentions the acknowledged limitation of relying on "information-rich query tokens" for saliency map computation with potentially vague or ambiguous queries. We first clarified that queries like "Describe this image," while general, are visually dependent, and our CHAIR dataset results for image captioning demonstrate robust performance even with general vision-dependent queries. In addition, we have elaborated concrete mitigation strategies for future work. Moreover, we identified the gap that no existing benchmark specifically targets scenarios where answering does not depend on visual information despite image input, or where substantial query portions are unrelated to image content. Creating such a dataset represents valuable future work for the broader research community.

**Reviewer 37D4**: Regarding performance gains on general benchmarks (MMStar and MMBench): we do not anticipate significant gains on these benchmarks, as errors typically stem from reasoning limitations rather than perceptual hallucinations. Our primary goal is to maintain reasoning capability while reducing hallucinations. In the original submission, we evaluated GIFT on two general benchmarks SEED-Bench and MME, showing performance comparable to greedy decoding with minimal impact on reasoning capabilities. In contrast, other baseline methods show mixed results, suggesting they face the inherent perception-reasoning trade-off. Our additional evaluation on MMBench and MMStar (the two datasets shared by the reviewer) further demonstrates that GIFT maintains comparable performance to greedy baseline across all models, validating its ability to preserve reasoning capabilities.

---

### Meta-Review · Area_Chair_Psx4 · 2026-01-02

**Summary:**

The initial score of the paper is 2446, and none of the reviewers changed their score before the re-assignment. After reading the paper and rebuttal, the AC trend to reject this paper due to the following problem:
1. Limited comparison with previous works.
2. Fine-grained hyper-parameter searching.
3. Results on new models. According to the rebuttal results, the Qwen3-VL shows significantly worse performance than its previous version ( Qwen2-VL) on both CHAIR_S and CHAIR_I, raising concerns about the correctness of the evaluation

**Reviewer Concerns:**

Please refer to the Summary

**Reviewer Scores:**

Till the deadline, none of the reviewers changed their score. According to the rebuttal, the AC thinks the reviewer 37D4 may raise the score

### No change
Reviewer iBdX and nbc4: One critical concern is the performance on new models, while the Qwen3-VL performance is abnormal and not convincing.
Reviewer aN7r: The rebuttal failed to solve the concern about visual reliance with solid results.

### Raise
Reviewer 37D4: The results on more benchmarks and methods are provided

---

### Decision · Program_Chairs · 2026-01-26

Reject